

# Observations of climatologically invariant scale-invariance describing cloud horizontal sizes

Thomas D. DeWitt[1], Timothy J. Garrett[1], Karlie N. Rees[1], Corey Bois[1], and Steven K. Krueger[1]

[1]Department of Atmospheric Sciences, University of Utah, 135 S 1460 E Rm 819, Salt Lake City, UT 84112, USA;

**Correspondence:** tim.garrett@utah.edu

**Abstract.**

The numbers of clouds of a given size is a defining feature of the earth's atmosphere. As well as cloud area, cloud perimeter $p$ is interesting because it represents the length of the shared interface between clouds and clear-skies across which air and buoyant energy are dissipated. A recent study introduced a first-principles expression for the steady-state distribution of cloud perimeters, measured within a quasi-horizontal moist isentropic layer, that is a scale invariant power-law $n(p) \propto p^{-(1+\beta)}$, where $n(p)$ is the number density of cloud perimeters within $[p, p+dp]$ and $\beta = 1$. This value of $\beta$ was found to be in close agreement with output from a high-resolution, large eddy simulation of tropical convection. To further test this formulation, the current study evaluates $n(p)$ within near-global imagery from nine full-disk and polar-orbiting satellites. A power-law is found to apply to measurements of $n(p)$, and the value of $\beta$ is observed to be remarkably robust to latitude, season, and land/ocean contrasts suggesting that, at least statistically speaking, cloud perimeter distributions are determined more by atmospheric stability than Coriolis forces, surface temperature, or contrasts in aerosol loading between continental and marine environments. However, the measured value of $\beta$ is found to be $1.29 \pm 0.05$ rather than $\beta = 1$, indicating a relative scarcity of large clouds in satellite observations compared to theory and high-resolution cloud modeling. The reason for this discrepancy is unclear but may owe to the difference in perspective between evaluating $n(p)$ along quasi-horizontal moist isentropes rather than looking down from space. As a test of this hypothesis, numerical simulation output shows that, while $\beta \sim 1$ within isentropes, higher values of $\beta$ are reproduced for a simulated satellite view. However, the simulated value is a function of the cloud detection sensitivity where little such sensitivity is seen in satellite observations suggesting a possible misrepresentation of the physics controlling cloud sizes in simulations. A power-law also applies to satellite observations of cloud areas covering a range between $\sim 3\,\mathrm{km}^2$ and $\sim 3 \times 10^5\,\mathrm{km}^2$, a much wider range of scales than has been previously described in studies that we argue inappropriately treated the statistics of clouds truncated by the edge of a measurement domain.

## 1 Introduction

Since the first numerical global climate models (GCMs) were developed in the 1960s, there have been exponential advances in computational capabilities that have led to spectacular simulations of cloud structures. The next generation of climate models is expected to resolve individual clouds at kilometer scales (Schär et al., 2020). The strategy behind this "bottom up" approach to determining the role of clouds in climate is that pursuing ever finer spatial resolution and improved model physics will lead to





more accurate predictions, accepting the necessary evil of increased computational expense (Slingo et al., 2022). Yet, perhaps alarmingly, it has not been clear that this approach has been successful in its goal given that the spread in GCM predictions of the climate sensitivity to greenhouse gases has, if anything, only increased (Arias et al., 2021; Palmer, 2016; Lovejoy, 2022).

In some sense, time-dependent deterministic simulations are not obviously well suited for obtaining a statistical time-independent climatology. An alternative approach might be to derive physical properties, using principles of statistical thermodynamics, from bulk physical constraints (Procyk et al., 2022; Arakawa, 2004). A familiar example is the simplicity of the derivation of the Maxwell-Boltzmann statistics characterizing the distribution of speeds of molecules in an ideal gas, obtained knowing only the average energy per molecule and without deterministically simulating individual particles and the extraordinary complexities of their quantum mechanical interactions (Schroeder, 2021). There is some evidence this "top-down" philosophy may work for convective cloud fields. An exponential distribution of mass fluxes can be derived for non-interacting clouds by considering only the large-scale vertical mass flux (Cohen and Craig, 2006; Craig and Cohen, 2006). A more recent study (Garrett et al., 2018) took a similar top-down approach but allowed for cloud interactions. It obtained a distribution of cloud horizontal sizes at steady-state, arguing they follow an exponential in saturated static energy and a power-law with respect to cloud perimeter.

In this study, we use a range of satellite observations to test the validity of the cloud perimeter distribution derived by Garrett et al. (2018). We show that both cloud perimeters and cloud areas do indeed follow a power-law, but that the power-law exponent appears to be a function of perspective, agreeing well with theory in thin horizontal layers in cloud-resolving models but not to satellite observations of cloud fields looking down from space. We also find that the choice of domain size and treatment of clouds that are truncated by the domain edge can introduce spurious scale breaks in power-law size distributions. We suggest previous results that do not account for these subtle effects should be interpreted with caution.

## 2   A steady-state thermodynamic model for cloud size distributions

To begin, we justify why it is suitable to look at cloud perimeters by summarizing the derivation of the cloud perimeter number distribution $n(p)$ presented by Garrett et al. (2018). The foundation is an idealized thermodynamic cycle around cloud edges – what was termed a "mixing engine" – defined by four "legs":

1. Moist adiabatic ascent inside cloud

2. Diabatic mixing of cloudy and clear air across cloud edge that dries the air and reduces cloud perimeter

3. Dry adiabatic clear-sky descent

4. Diabatic mixing of cloudy and clear air across cloud edge that moistens the air and increases cloud perimeter

The cycle is analogous to the familiar Carnot cycle, used to describe hurricanes (Emanuel, 1991), but with entropy generation associated with mixing at cloud edge rather than with energetic exchanges with the oceans or outer space. In observations of tropical convection, Heus and Jonker (2008) found that shallow cumulus tend to have a neutrally buoyant cloud edge and





a "subsiding shell" of descending clear air adjacent to cloud edge. A similar pattern was later observed in local circulations around simulated deep convection (Glenn and Krueger, 2014). These observations appear to support the mixing engine framework, at least for actively convecting clouds.

Representing mass fluxes across cloud edges in 4-D space-time coordinates, as is typically done in detailed cloud numerical simulations, is difficult because turbulent mixing changes both the location and length of the cloud edge itself, and over a very wide range of time and space scales. However, while cloud edge may deform during mixing, it maintains its position as a point of approximate neutral buoyancy, and in this sense can serve as a fixed reference point in a related coordinate system. For this purpose, we use the moist static energy, which is given by

$$h = gz + c_p T + L_v q,\tag{1}$$

where $g$, $c_p$, and $L_v$ are the gravitational acceleration, the specific heat of air at constant pressure, and the latent heat of vaporization of water, respectively, and $z$, $T$, and $q$ are height, temperature, and the mixing ratio, respectively. At cloud edge, air is just saturated, so the moist static energy is equal to the *saturated* static energy, defined as $h^\star = h(q = q^\star)$ where $q^\star$ is the saturated mixing ratio. At a given height, perturbations in saturated static energy can be related to temperature (and hence buoyancy) perturbations $T'$ through $h' = c_p(1 + \gamma)T'$ where $\gamma = L/c_p dq^\star/dT$ (Randall, 1980).

In a convectively unstable atmosphere, variability in $h^\star$ is dominated by its variability with respect to height, so a constant $h^\star$ surface can be approximated as lying along a surface of constant $z$ (Xu and Emanuel, 1989). Supposing a thin atmospheric layer of thickness $\delta z$, clouds within this layer can be partitioned into discrete bins $j$ of mean perimeter $p_j$. For a number $n_j$ of such clouds, each bin has a total cloud perimeter $n_j p_j$ and a total surface area normal to the horizontal $\sigma = n_j p_j \delta z$. Fick's law suggests that for bin $j$, the total rate of dissipation of potential energy across cloud edge $Q_j$ due to diabatic turbulent mixing is proportional to the product of the energy gradient between cloudy and clear air $\nabla h$ and the total surface area $\sigma$. Provided the perturbation from the domain mean $\delta h$ is much smaller than the mean value $\langle h^* \rangle$, a constraint that is always satisfied in the troposphere, and that circulations around cloud edge area are approximately isotropic (Heus and Jonker, 2008; Heus et al., 2009; Wang et al., 2009), $\nabla h \approx \delta h/\delta z = S$ where $S$ is the stability. Thus the horizontal rate of dissipation of energy due to turbulent mixing across cloud edge in any given size bin $j$ is

$$Q_j \propto n_j p_j \delta h.\tag{2}$$

In any cloud field, clouds continually grow and shrink due to turbulent mixing processes, and so cloud number is passed from one perimeter bin $j$ to the next $j+1$ or $j-1$. At steady-state, however, which can be defined as a time invariant perimeter distribution, there must be no net convergence of cloud number or energy into any bin $j$. This implies that $dQ/dp = 0$, or in discretized form from Eq. 2, that $n_j p_j = const$. The steady-state perimeter distribution $n(p)$ can therefore be expected to follow the power-law (Garrett et al., 2018)

$$n(p) \equiv \frac{dn}{dp} \propto p^{-(1+\beta)}, \qquad \beta = 1, \qquad p_{\min} < p < p_{\max}.\tag{3}$$

The continuous function $n(p)$ can be discretized into linearly spaced bins with constant $\Delta p$, in which case the slope on a plot with two logarithmic axes would be $-(1 + \beta)$. If logarithmically-binned, with constant $\Delta \ln p$, the slope of the power-law



would be $-\beta$ because $dn/d\ln p = pn(p)$. We favor logarithmically-spaced bins as being better suited to describe the vast range of cloud sizes because linearly-spaced bins can be associated with poor sampling in large bins where a power-law is anticipated (Clauset et al., 2009).

## 3   The challenge of measuring cloud size distributions

There is an inevitable conundrum of how to best measure the property of scale-invariance with a finite domain. Power-laws

such as Eq. 3 are generally considered to be "scaling", since a rescaling of $p$ by some constant factor $c$ results in a constant rescaling of $n(p)$ by a constant factor $c^{-(1+\beta)}$. Of course, it is impossible for any physical system to exhibit scale invariance over an infinite range of scales, and so such scaling behavior can only be valid over a finite range $p_{\min} < p < p_{\max}$. Beyond these "scale breaks", the value of $\beta$ changes or the functional form of the distribution changes. As an example, a common feature of power-law distributions describing many other natural and social systems is an exponential cutoff (Newman, 2005;

Clauset et al., 2009).

For clouds, as a guess, the smallest possible size defining $p_{\min}$ might be the Kolmogorov microscale for turbulent circulations such as in the mixing engine, whose order of magnitude is $\sim 1\,\mathrm{mm}$ (Tennekes and Lumley, 1972). The largest possible clouds are of course limited by the Earth's circumference of $\sim 10^5\,\mathrm{km}$, but might be more reasonably constrained by the Rossby radius of deformation $\sim 10^3\,\mathrm{km}$ where Coriolis forces limit horizontal spreading.

Since cloud edges are fractal (Lovejoy, 1982), calculated perimeter lengths depend on the chosen measurement resolution, so $p_{\max}$ can be orders of magnitude larger than the distance from one end of a cloud to the other. A measure of maximum cloud size that is less resolution-dependent is maximum cloud area $a_{\max}$, which is roughly $\mathcal{O}(\text{cloud length})^2$.

For the power law exponent, Garrett et al. (2018) found $\beta = 1.06 \pm 0.02$ in a comparison with a highly detailed numerical simulation of a tropical cloud field, in close agreement with the theoretically expected value of $\beta = 1$. Cloud perimeter distri-

butions have yet to be assessed observationally although cloud area distributions have been widely studied, generally revealing power-law distributions in both satellite observations (Wood and Field, 2011; Koren et al., 2008; Benner and Curry, 1998; Cahalan and Joseph, 1989; Kuo et al., 1993) and models (Yamaguchi and Feingold, 2013; Neggers et al., 2003, 2019; Christensen and Driver, 2021), although not in every study (López, 1977). Assuming both cloud areas and perimeters are power-law distributed, the two quantities can be related by the scaling relationship

$$p = const. \times a^{D/2}. \tag{4}$$

$D$ is often interpreted to be the fractal dimension $D_f$ as it is formally defined by the relation $l \propto \xi^{1-D_f}$ relating how the measured length $l$ of a fractal line such as cloud perimeter depends on the "ruler length" (or resolution) $\xi$ used to measure it (Mandelbrot, 1982). Assuming the relationship $D = D_f$ is valid, the fractal dimension can be determined by fitting a linear regression between observations of $\ln \sqrt{a}$ and $\ln p$ (e.g. Lovejoy, 1982; Cahalan and Joseph, 1989; Christensen and Driver,

2021; Siebesma and Jonker, 2000).

Subsequent work has shown that for clouds $D$ is not in fact strictly equivalent to the fractal dimension $D_f$. Batista-Tomás et al. (2016) and Peters et al. (2009) pointed out that adopting the equivalence $D = D_f$ requires holes in clouds to be excluded



from contributing to the cloud's perimeter, as the fractal dimension is a property of a single curve (a cloud's exterior perimeter) rather than an ensemble of curves (a cloud's exterior perimeter, and the perimeter of each hole).

Furthermore, Imre (1992) showed that the constant pre-factor in Eq. 4 often itself scales with $a$. In this case, fitting a regression line to a scatterplot of $\ln\sqrt{a}$ vs. $\ln p$ would yield a value for $D$ that implicitly includes a scaling contribution from the supposed "constant".

Setting aside these details, a scaling of the form Eq. 4 nonetheless can be used to empirically relate cloud areas and perimeters, making the expression useful regardless of any particular interpretation of $D$. Here, it permits the perimeter size distribution Eq. 3 to be converted to a distribution in cloud area. Since $d\ln a \propto d\ln p$, $dn/d\ln a \propto dn/d\ln p$, and so

$$n(a) \equiv \frac{dn}{da} \propto a^{-(1+\alpha)}, \qquad \alpha = \frac{D\beta}{2}, \qquad a_{\min} < a < a_{\max}. \tag{5}$$

Adopting $D \approx 4/3$ (Lovejoy, 1982; Siebesma and Jonker, 2000), noting that both higher (Christensen and Driver, 2021; Cahalan and Joseph, 1989) and lower (Batista-Tomás et al., 2016; Cahalan and Joseph, 1989) values have been measured, and $\beta = 1$ as proposed by (Garrett et al., 2018), then Eq. 5 yields $\alpha \approx 2/3$. By contrast, widely conflicting values are observed

for $\alpha$, as well as for the location of the scale break $a_{\max}$. Cahalan and Joseph (1989) and Benner and Curry (1998) found in satellite observations values for $\alpha$ of 0.89 and 0.98, respectively, and values for $a_{\max}$ ranging from $4\,\mathrm{km}^2$ to between $0.28\,\mathrm{km}^2$ and $0.62\,\mathrm{km}^2$, respectively. In large eddy simulations, Neggers et al. (2003) found that $\alpha = 0.70$, with scale breaks between $0.16\,\mathrm{km}^2 \leq a_{\max} \leq 1.6\,\mathrm{km}^2$.

For larger domains considered in other studies $a_{\max}$ tends to be larger. Wood and Field (2011) found using MODIS satellite

data that $\alpha = 0.87 \pm 0.03$ and $a_{\max} \gtrsim 10^6\,\mathrm{km}^2$. Peters et al. (2009) found a scale break in mesoscale convective clusters at $a_{\max} \sim 10^5\,\mathrm{km}^2$, although it depended on the value of a threshold based on column water vapor. Conversely, Christensen and Driver (2021) found $a_{\max} \sim 10^6\,\mathrm{km}^2$ for tropical deep convection. There is also variation in calculated values for $\alpha$, as Koren et al. (2008) and Yamaguchi and Feingold (2013) found $\alpha = 0.3 \pm 0.1$ and $\alpha = 0.59$, respectively, and no evidence for a scale break $a_{\max}$, although they considered smaller domains.

These conflicting results could reflect meteorological differences, as there is some evidence $D$, and therefore $\alpha$ through Eq. 5, is itself dependent on cloud type and size (Batista-Tomás et al., 2016; Cahalan and Joseph, 1989). However, a largely overlooked explanation for the surprising variance in values for $\alpha$ and $a_{\max}$ is one of sampling bias. Larger clouds are more likely to be truncated by the edge of the measurement domain than small clouds, and if they are removed from the analysis, as is sometimes done, there can be a spurious scale break introduced to the size distribution. Such a scale break would depend

only on the size of domain considered, rather than some intrinsic physical property of the cloud field itself.

Past studies generally do not mention how clouds truncated by the domain edge are treated or, in some cases, they simply remove them from analysis (e.g. Christensen and Driver, 2021; Peters et al., 2009). Plausibly, some of the inconsistencies seen in measured values of $a_{\max}$ and $\alpha$ could owe to this measurement problem. For example, one seeming solution is to retain clouds that are truncated by the domain edge but measure only the portion of the cloud area that lies within the domain. With

this approach, a portion of the given cloud's area is necessarily omitted, likely placing it in a smaller size bin where counts are consequently oversampled.



## 4 Methods

Our goal here is to test in satellite observations and models the hypotheses proposed by Garrett et al. (2018), namely that $\beta = 1$ as specified by Eq. 3 and that the value of $\beta$ is the same for any cloud field at steady-state. Second, we attempt to address

inconsistencies in previous observations of $\alpha$ and $a_{\max}$ by appropriately accounting for bias introduced by the treatment of clouds truncated by the edge of a satellite measurement domain.

### 4.1 Satellite datasets

The satellite platforms used to image clouds in this study fall into two broad categories: full-disk and polar-orbiting. Full-disk images are effectively a snapshot of Earth taken from geostationary orbit or, in the case of EPIC, the L1 Lagrange point. Polar-

orbiting sensors continuously scan a rectangular swath as they move poleward. Details about the datasets are summarized in Table 1.

**Table 1.** Satellite datasets used in this study.

| Sensor Name | Satellite | View Type | Approx. nadir resolution | Longitude at nadir | Dates examined | Description of cloud mask algorithm |
|---|---|---|---|---|---|---|
| **GOES WEST** | GOES-17 | Full-Disk | 2 km | 137° W | 01 January 2021 to 01 January 2022 | Derrien and Gléau (2005, 2010) |
| **GOES EAST** | GOES-16 | Full-Disk | 2 km | 75° W | 01 January 2021 to 01 January 2022 | Derrien and Gléau (2005, 2010) |
| **METEOSAT 11** | METEOSAT 11 | Full-Disk | 3 km | 0° | 01 January 2021 to 01 January 2022 | Derrien and Gléau (2005, 2010) |
| **METEOSAT 9** | METEOSAT 9 | Full-Disk | 3 km | 42° E | 01 January 2021 to 01 January 2022 | Derrien and Gléau (2005, 2010) |
| **Himawari** | Himawari | Full-Disk | 2 km | 141° E | 01 January 2021 to 01 January 2022 | Derrien and Gléau (2005, 2010) |
| **EPIC** | DISCOVR | Full-Disk | 8 km | - | 01 January 2017 to 01 January 2018 | Yang et al. (2019) |
| **VIIRS** | NOAA-20 | Polar-Orbiting | 750 m | - | 01 January 2021 to 01 January 2022 | Kopp et al. (2014) |
| **POLDER** | PARASOL | Polar-Orbiting | 1/18° | - | 01 January 2012 to 01 January 2013 | Buriez et al. (1997) |
| **MODIS 1km** | TERRA | Polar-Orbiting | 1 km | - | 01 January 2012 to 01 January 2013 | Ackerman et al. (1998, 2008) |
| **MODIS 250m** | TERRA | Polar-Orbiting | 0.25 km | - | 01 January 2012 to 10 January 2012 | Section 4.1.2 |





### 4.1.1 Pre-processed cloud masks

For most of the satellite datasets described in Table 1, individual clouds are identified from pre-processed binary cloud masks designed to distinguish cloudy and clear sky. The definition of a cloud is somewhat subjective, and so inevitable differences in cloud identification algorithms and sensor capabilities lead to variations in global cloud coverage estimates between datasets. Even for a given satellite dataset, estimates of global cloud fraction depend on the choice of viewing angle, increasing with more oblique perspectives (Maddux et al., 2010). To mitigate this concern, images are truncated to exclude cloud imagery where the sensor zenith angle is greater than $60°$, a choice intended as a compromise between limiting sensitivity to viewing angle while retaining a large domain area.

### 4.1.2 Cloud masks based on reflectance thresholds

To test the sensitivity of measured distributions of cloud sizes to cloud definition, we use a simple cloud mask based on MODIS band 1 optical reflectance $R$, which is sensitive to wavelengths between $620\,nm$ and $670\,nm$ and has a resolution at nadir of $250\,m$. We examine 13 tropical maritime granules, each centered between approximately $10°S$ and $20°N$ and $115°W$ to $140°W$ and covering an area approximately $1950\,km$ wide by $2030\,km$ long. Images from each granule were visually inspected for artifacts from sun glint, and several additional granules were omitted from the analysis due to sun glint contamination. Figure 1 compares several example cloud masks generated using various thresholds in $R$ alongside the pre-processed cloud mask and an RGB image.

### 4.2 SAM numerical simulations

For numerical simulations of cloud fields we use output from the System for Atmospheric Modeling (SAM) (Khairoutdinov and Randall, 2003). SAM was initialized and forced by large-scale thermodynamic tendencies derived from mean conditions during the GATE Phase III field experiment (Khairoutdinov et al., 2009) and run with prescribed radiative heating, diagnostic subgrid-scale turbulence, and two prognostic hydrometeor variables (precipitating and non-precipitating) from which cloud water, cloud ice, rain, snow, and graupel are diagnosed. The simulation's domain size is $204.8\,km \times 204.8\,km$ with $100\,m$ horizontal grid spacing and a $2\,s$ time step. The vertical grid spacing is $50\,m$ below $z = 1.2\,km$ and increases to $100\,m$ at $z = 5\,km$. There are a total of 210 vertical levels.

Shallow cumulus form in the first hour of the simulation, gradually deepening into deep convection by hour 6 with a steady-state reached beyond approximately hour 12. We analyze hourly 3-D model output from hours 12 to 24. Output from this simulation was also used in Garrett et al. (2018) and is described in full detail in Khairoutdinov et al. (2009).

A cloud mask for each horizontal layer in the simulation was applied by setting all grid cells with non-precipitating cloud condensate mixing ratios $q_n$ (including both liquid and ice) in excess of 1% of the saturated mixing ratio $q^\star$ to cloudy and the remainder to clear. Once every grid cell is defined as either cloudy or clear, 2-D images were created by isolating each individual height level in the domain, creating 210 images for every time step. These images were then analyzed separately using the same method as the satellite imagery (described below). Isolating individual horizontal layers in this manner provides





**Figure 1.** Example image, pre-processed cloud mask, and cloud masks created from various thresholds in optical reflectance $R$ for a single MODIS granule. In the reflectance-based cloud masks, pixels with reflectance higher than the threshold are set to cloudy (white), while the others are set to clear (dark blue). The image is centered at approximately 1° S, 130° W and was taken at approximately 01 January 2021, 19:05 UTC. Note that pixels are depicted here as being uniform in size but that cloud size calculations account for pixel size increasing away from nadir.





an approximate method of isolating constant $h^\star$ surfaces (Garrett et al., 2018). After perimeters were calculated and binned for
each layer, counts were summed over all layers.

### 4.3   Cloud identification and filtering

Both the satellite and model datasets yield 2-D binary cloud masks where grid cells or pixels are either cloudy or clear.
Individual clouds are defined as connected cloudy regions, identified by applying a convention that adjacent cloudy pixels are
connected whereas diagonal cloudy pixels are not (termed "4-connectivity"; Wood and Field (2011); Christensen and Driver
(2021)), although the analysis is not significantly affected by this choice (Kuo et al., 1993). In the satellite datasets, the pixel
lengths in the $x$ and $y$ directions are determined independently as a function of satellite distance and sensor zenith angle.

     Cloud perimeter is then computed by summing all pixel side lengths along the edge of the cloud, and cloud area by summing
the areas of each individual cloudy pixel. Cloud holes add to the clouds' perimeter but reduce its area, which as described
above implies $D \neq D_f$ in Eq. 4. Clouds consisting of a small number of pixels are more Euclidean than fractal (Christensen
and Driver, 2021), which leads to an inaccurate estimate of the small portion of the size distribution. We therefore truncate
number distributions to exclude cloud perimeters $\leq 10\times$(resolution at nadir) or areas $\leq 10\times$(resolution at nadir)$^2$. The smaller
portion of the size distributions obtained using EPIC display non-power-law behavior, in contrast to all other satellite datasets
over similar scales, and therefore we exclude EPIC clouds with perimeters $\leq 30\times$(nadir resolution) or areas $\leq 1000\times$(nadir
resolution)$^2$ (see Appendix A for further discussion).

To account for possible scale breaks in a size distribution introduced by clouds truncated by the edge of a measurement
domain, area or perimeter bins in which the number of clouds truncated by the edge is greater than 50% of the total in that bin
are removed from consideration. For the observed cloud fields, such bins tend to be those in the larger end of the size spectrum
as large clouds are most likely to touch the domain edge. The threshold choice of 50% represents a compromise, removing
bins most sensitive to border effects while allowing for a large range of cloud sizes to be studied. Calculated values for $\beta$ and
$\alpha$ are relatively insensitive to more stringent thresholds less than 50%.

     We calculate the power-law exponents $\alpha$ and $\beta$ by performing a linear regression in logarithmic space since, from Eqs. 3
and 5, $\ln n(p) = -\beta \ln p + const.$ and $\ln n(a) = -\alpha \ln a + const.$ It has been argued this method can lead to underestimates
of the exponent (Clauset et al., 2009), but there is no straightforward alternative when presented, as is the case here, with a
power-law that has both an upper and lower bound (Hanel et al., 2017). We evaluate uncertainties as 95% confidence inter-
vals corresponding to two standard errors of the regression. We report numerical values of plotted data in the Supplemental
Information.

## 5   Measured cloud size distributions

In satellite observations, both cloud areas and perimeters are well described by a power-law distribution. For cloud areas, Fig.
2 shows measured values of $\alpha$ ranging between $\alpha = 0.90 \pm 0.02$ (POLDER and MODIS 250m) and $\alpha = 0.99 \pm 0.02$ (GOES





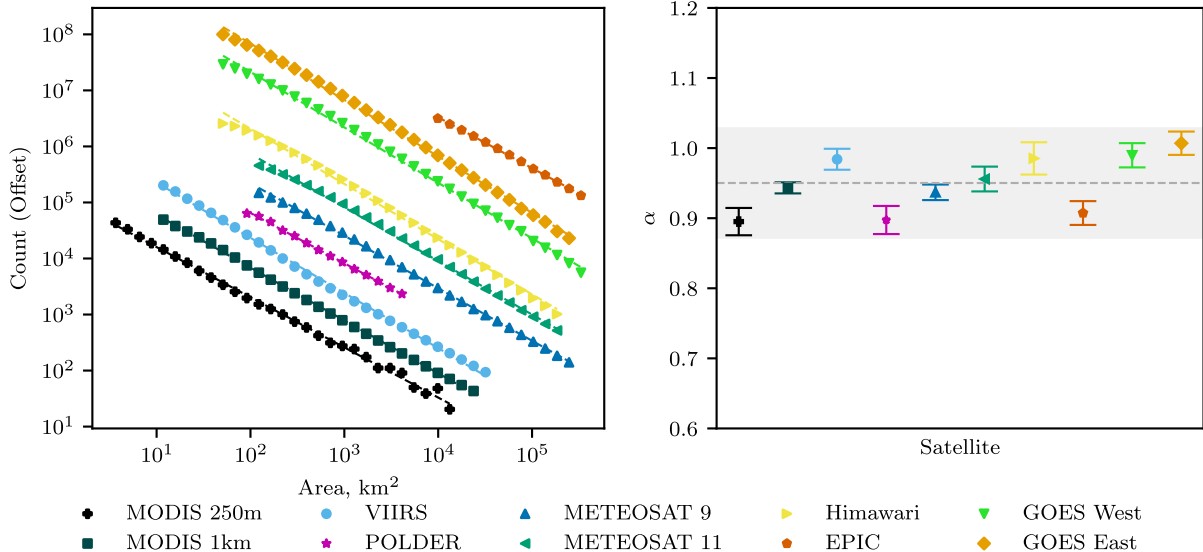

**Figure 2.** Left: Logarithmically-binned histograms of cloud areas for satellite datasets. Right: Measured values of the power-law exponent $\alpha$ (Eq. 5) with associated mean (gray line) and 95% confidence interval (gray box). Counts have been vertically offset for clarity. Uncertainties represent 95% confidence intervals, derived from a linear regression standard error analysis.

East and Himawari) and a mean value, across all satellite datasets, of $\langle\alpha\rangle = 0.95 \pm 0.08$. These values are largely in agreement with several previous studies (e.g. Wood and Field (2011), Cahalan and Joseph (1989), and Benner and Curry (1998)).

For cloud perimeters, Fig. 3 shows values of $\beta$ ranging from $\beta = 1.22 \pm 0.02$ (MODIS 1km and 250m) to $\beta = 1.316 \pm 0.008$ (GOES East), with a mean across all satellite datasets of $\langle\beta\rangle = 1.26 \pm 0.06$. This value differs from that found in SAM horizontal levels, where $\beta = 0.98 \pm 0.03$, and from the theoretically-derived value of $\beta = 1$ (Eq. 3).

These mean values of $\alpha$ and $\beta$ imply, from Eq. 4, that $D = 1.5 \pm 0.1$, which is in good agreement with prior studies that have generally found values of $D$ slightly greater than 4/3, e.g. $D = 1.35$ (Lovejoy, 1982), $1.25 \leq D \leq 1.59$ (Cahalan and Joseph, 1989), or $D = 1.4$ (Christensen and Driver, 2021).

After omitting bins containing 50% or more clouds truncated by the domain edge, a scale break $a_{\max}$ is no longer evident in the area distributions. In several cases, the distributions exhibit scale invariance extending to areas larger than $10^5\,\mathrm{km}^2$, with the largest to at least $\sim 3 \times 10^5\,\mathrm{km}^2$ (EPIC). We find that $a_{\max}$ must therefore have a value larger than roughly $3 \times 10^5\,\mathrm{km}^2$, corresponding to an effective diameter of $\sim 600\,\mathrm{km}$, substantially larger than some have previously suggested (e.g. Cahalan and Joseph, 1989; Benner and Curry, 1998; Neggers et al., 2003), with Wood and Field (2011) extending $a_{\max}$ to $10^6\,\mathrm{km}^2$.

## 5.1 Variability with seasonality, latitude, and surface type

The perimeter size distribution, given by Eq. 3, was derived without explicit consideration of local climatological characteristics such as as season, surface type, or latitudinal location (Garrett et al., 2018). In satellite observations, the sensitivity of $\beta$ to





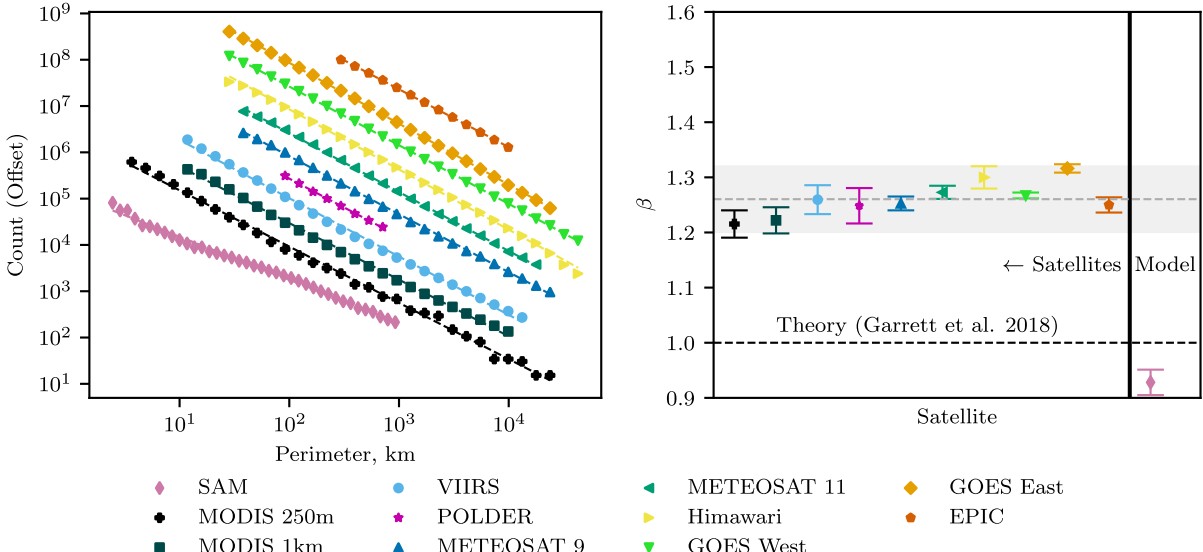

**Figure 3.** As in Fig. 2, but considering cloud perimeters (Eqn 3) and including results from SAM horizontal levels. The grey line and box on the right plot represent the mean and 95% confidence interval, respectively, across only the satellite datasets (excluding SAM).

such considerations is shown in Figs. 4, 5, and 6. Comparisons between latitude bands shown in Figs. 4 and 6 are restricted to observations using the polar-orbiting satellites MODIS and VIIRS because imagery from these sensors, regardless of latitudinal location, is both similar in domain area and always centered directly below the satellite. These conditions reduce the likelihood of bias due to differing viewing geometry, and neither condition holds for the full disk images. POLDER has been omitted from

Figs. 4, 5, and 6 because, when limited to smaller domains, its smaller sample size introduces significant statistical variability.

Measured values of $\beta$ appear robust across latitudinal regions, land/ocean contrast, and seasons, independent of sensor. Fig. 4 does show modest variability in the value of $\beta$ by month in the midlatitude regions 60°S-30°S and 30°N-60°N between a minimum value of $\beta = 1.21 \pm 0.03$ (MODIS, March, May, June, northern midlatitudes) and a maximum value of $1.32 \pm 0.02$ (MODIS, June, July, southern midlatitudes). Annual mean values of $\beta$ for the midlatitude and equatorial regions in Fig. 6

show similar values ranging from $\beta = 1.22 \pm 0.03$ for MODIS at northern midlatitudes to $\beta = 1.28 \pm 0.03$ for VIIRS in all regions and MODIS at southern midlatitudes. Separating clouds by marine and continental regions in Fig. 5, mean values for $\beta$ are $1.25 \pm 0.05$ for land and $1.28 \pm 0.04$ for ocean. All values are consistent with the global mean value across datasets of $\langle \beta \rangle = 1.26 \pm 0.06$.

## 6    Discussion

After accounting for spurious scale breaks introduced by the problem of attempting to measure the extent of scale invariance with a finite domain, we find that a power-law describes the distributions of both cloud perimeters and areas for a size range



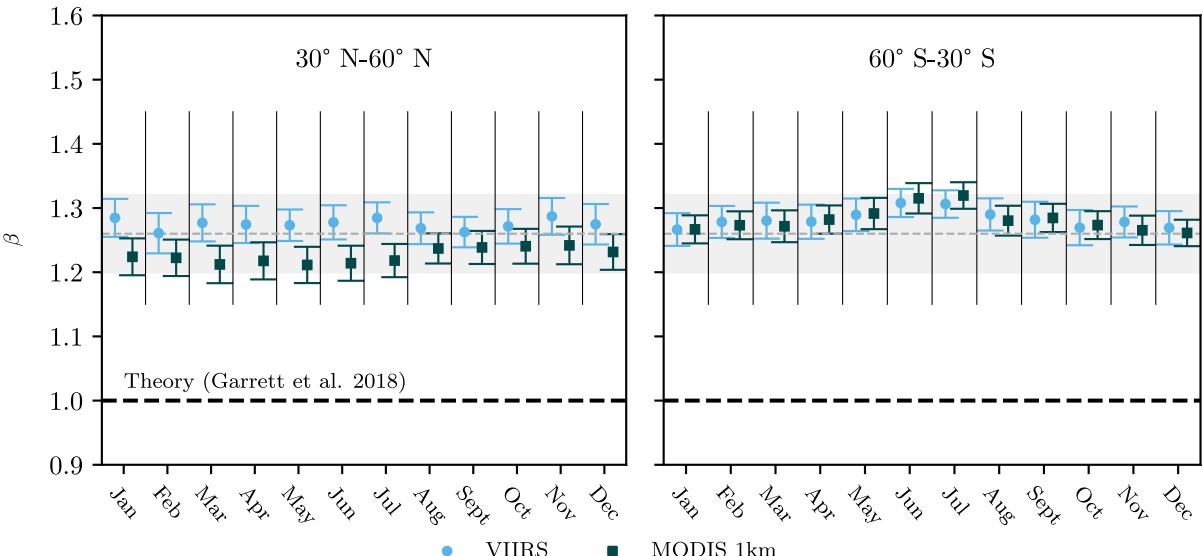

**Figure 4.** Measured values of $\beta$ (Eq. 3), for the northern midlatitude region (left) and the southern midlatitude region (right), separated by month. The equatorial region (not shown) shows similar variability around a mean value shown in Fig. 6.

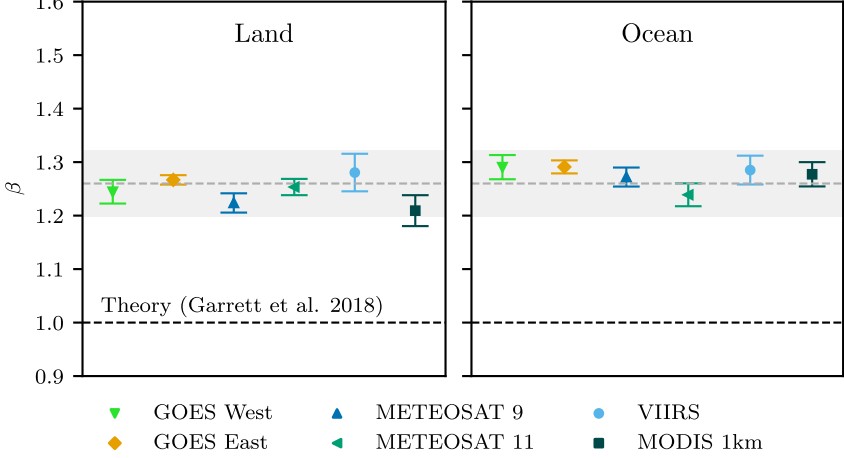

**Figure 5.** Measured values of $\beta$ (Eq. 3), separated into clouds over land (left) and clouds over water (ocean).



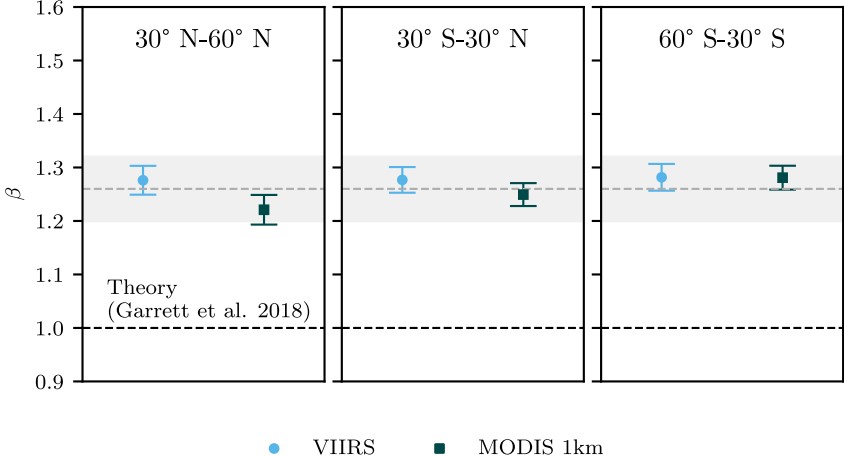

**Figure 6.** Measured values of $\beta$ (Eq. 3), separated into the northern midlatitude region (left), the equatorial region (middle), and the southern midlatitude region (right).

spanning four and five orders of magnitude, respectively, and likely extends even further. This result is perhaps all the more remarkable for the fact that the value of the exponent $\beta$ appears to be robust to such local climatological characteristics as season, latitude, land/ocean contrasts, or latitude as might be related to surface temperature, the Coriolis force, dominant cloud

type, or aerosol loading. In this sense, the observations appear to lend support to the general theoretical "mixing-engine" approach employed by Garrett et al. (2018) to obtain Eq. 3, where $\beta$ was derived only by considering mixing processes at cloud edge.

However, a puzzle remains: the global mean value of $\langle \beta \rangle = 1.26 \pm 0.06$ in satellite observations is higher than the value of $\beta \simeq 1$ obtained both theoretically (Eq. 3) and from SAM model simulations (Fig. 3). The difference is significant given

the range of scales in cloud sizes involved. For example, for a roughly three order of magnitude measured range for cloud perimeters, the discrepancy would imply an order of magnitude difference in cloud counts.

The most obvious inference is that the theory is missing something fundamental about what determines cloud perimeters, even if it produced perimeter distribution values of $\beta$ very close to those seen in a highly detailed numerical cloud model. Alternatively, one important distinction that may be made between the two approaches is simply one of perspective. Perimeter

distributions from the numerical model SAM shown in Fig. 3 and previously in Garrett et al. (2018) were calculated by treating every individual horizontal layer in the SAM volume as an independent 2D image. Only after each cloud perimeter was calculated and binned were the counts summed over all layers to create a single histogram, with no account made for cloud overlap. We term this method "layers".

Satellite imagery differs, as it offers a two-dimensional representation of a cloud field as seen from above rather than within.

Any vertical cloud overlap is effectively "compressed" into a single horizontal plane, *before* individual cloud perimeters are calculated. No distinction is made between overlapping clouds and vertically continuous clouds.





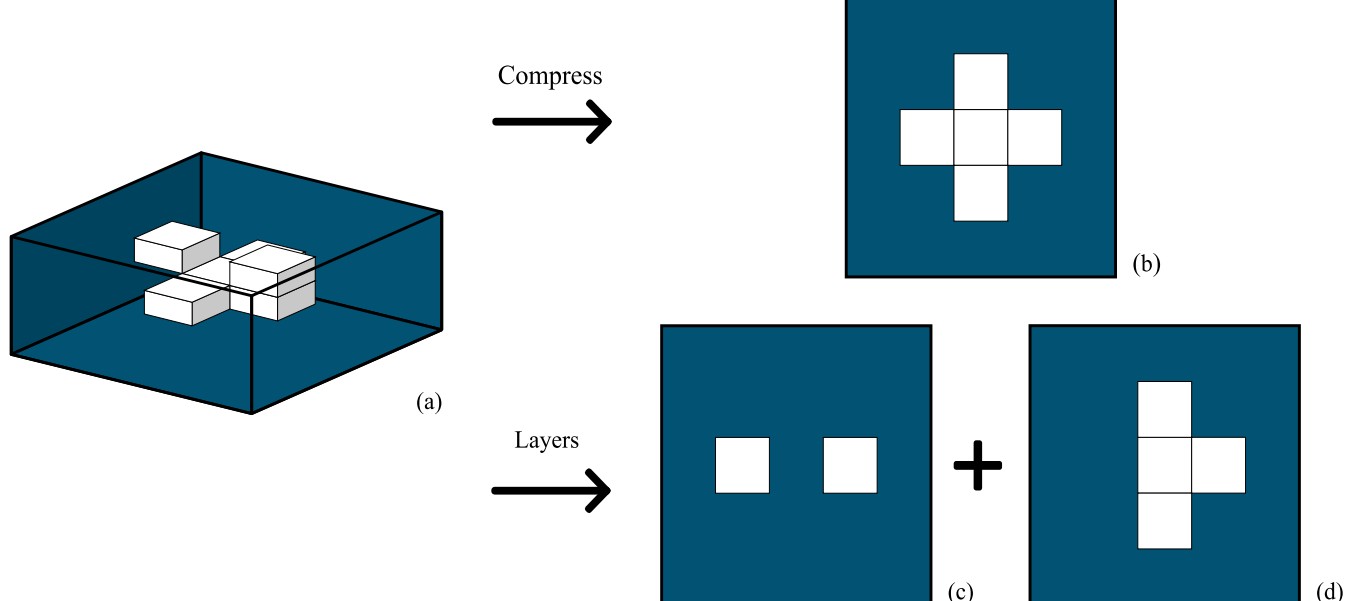

**Figure 7.** Example comparison of two methods of measuring perimeters of a 3-D cloud in SAM. The "compress" method creates a 2-D image by vertically summing cloud properties, resulting in a single image for each volume representing how clouds might be seen from above. In contrast, the "layers" method used in Garrett et al. (2018) considers each horizontal slice as a separate image such that for $n$ horizontal layers, $n$ individual images would be produced and analyzed as independent images. In the example, the "compressed" method would produce one cloud with $p = 12$ pixels and the "layers" method would produce three clouds, one with $p = 10$ and two with $p = 4$.

For example, the idealized cloud field in Fig. 7 yields a single cloud with $p = 12$ in the compressed satellite view (b), whereas a layers analysis would see three clouds, two in layer (c), each with $p = 4$, and one in layer (d), with $p = 10$. A priori, we might therefore expect a compressed image to yield relatively fewer small clouds than the layers case, as is the case in the example. This would result in a smaller value of $\beta$ for the compressed case relative to the layers case. Counterintuitively, however, the opposite appears true: the value of $\beta$ is larger in the compressed satellite perimeter distribution than in the layered SAM distribution.

The difference between the two perspectives can be manufactured in SAM by creating vertically compressed images as they might be seen by a satellite from above. Here, this is accomplished by creating a 2D vertically-summed optical depth ($\tau$) field, to which a range of optical depth thresholds is applied to create a selection of cloud masks. Once binary cloud masks are created, clouds are identified and analyzed as described in Sect. 4.3.

Fig. 8 shows that values of $\beta$, calculated using the "layers" method in SAM, are consistent with the theoretical prediction $\beta = 1$ regardless of the threshold used to define cloud, but that the value of $\beta$ does indeed increase when the perspective is switched to one in which the clouds are vertically compressed as they might be seen from space. However, as the compressed threshold in $\tau$ grows, $\beta$ decreases, reaching a value of roughly 1 at $\tau \gtrsim 10$.



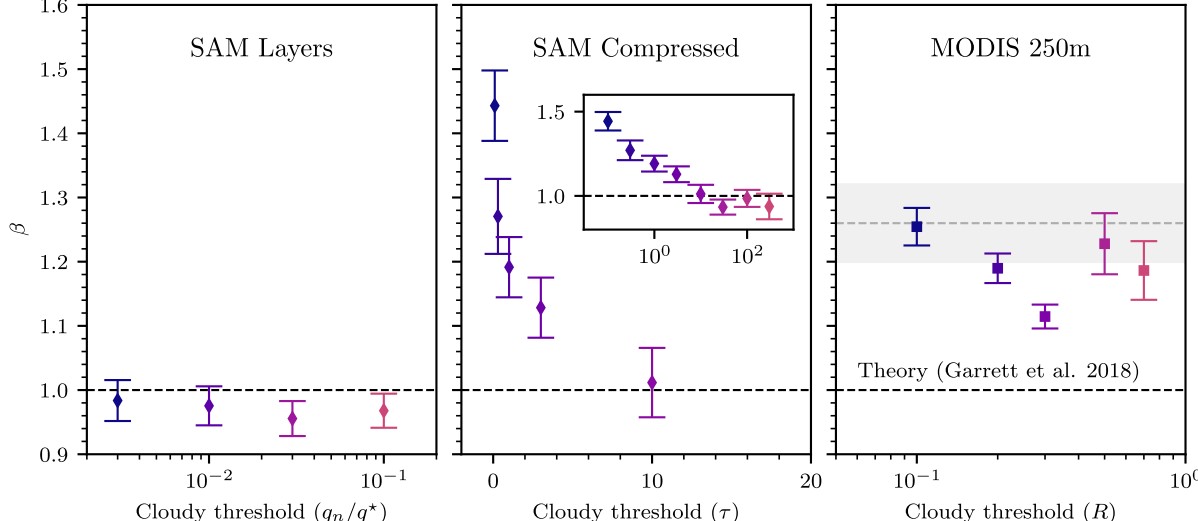

**Figure 8.** Left: Measurements of $\beta$ for individual horizontal layers ("layers") in SAM for varying thresholds in total cloud condensate $q_n$, normalized by the saturated mixing ratio $q^\star$. Middle: Measured values of $\beta$ for "compressed" images in SAM for varying model $\tau$ thresholds, created by vertically summing modeled $\tau$. The middle inset displays $\beta$ vs. $\tau$ for the compressed SAM data using a logarithmically scaled abscissa. Right: Measured values of $\beta$ for cloud masks of varying reflectance ($R$) thresholds for MODIS 250m data. See Figure 7 for a visualization of the difference between "compressed" images and "layers". The grey line and box indicate the global mean $\langle\beta\rangle = 1.26 \pm 0.06$ (Figure 3). Histograms from which values for $\beta$ are calculated are shown in Appendix **??**.

Such sensitivity of $\beta$ to optical depth threshold is at odds with observations, given that cloud masks specified by thresholds in reflectance between $R = 0.1$ and $R = 0.7$ for MODIS 250m data show very little trend in calculated $\beta$. Note, for comparison, that the range of reflectance thresholds considered is roughly equivalent to a range of optical depths between $\tau = 1$ and $\tau = 10$, and that the reflectance thresholds in MODIS 250m data generally produce values of $\beta$ that are consistent with the global mean value derived from the pre-processed MODIS cloud mask.

## 7 Conclusions

By considering cloud edges as a surface across which cloudy and clear skies compete for available convective potential energy through small scale mixing processes in a "mixing-engine", Garrett et al. (2018) derived a cloud perimeter distribution that follows a power-law $n(p) \propto p^{-(1+\beta)}$, where $\beta = 1$ (Eq. 3), for perimeters evaluated within thin isentropic layers. The prediction is independent of such considerations as the details of cloud microphysics or climatological state. We find in a detailed numerical simulation of a tropical cloud field that $\beta = 0.98 \pm 0.03$, which is consistent with the prediction.

In a wide range of satellite observations, however, the picture is more nuanced. Within measurement uncertainty, values of $\beta$ are insensitive to zonal band, land/ocean contrasts, or season, conditionally supporting the small-scale mixing engine





hypothesis. However, the globally-averaged value across all satellite datasets is significantly higher than predicted by either
theory or model with a value of $\beta = 1.26 \pm 0.06$.

The discrepancy likely owes to a difference in perspective between cloud size distributions measured within individual quasi-horizontal moist isentropic layers, as done for the numerical simulation, and those seen looking from above, as calculated from satellite observations. The precise explanation remains a puzzle, although we do see that values of $\beta$ are higher in numerical simulations when the perspective is changed to one looking from above, with clouds defined by a threshold in vertically summed optical depth. This may seem to help resolve the matter. But even here the picture is unclear since $\beta$ approaches unity as the optical depth threshold increases, and there is no such sensitivity to threshold seen in MODIS observations.

Regardless of the role of perspective, scale invariance appears to be a defining feature of clouds over at least four orders of magnitude in perimeter and five in area. We find that, in satellite observations, the limit of scale invariance in cloud area distributions $a_{\max}$ has a value larger than $3 \times 10^5\,\mathrm{km}^2$, a scale much larger than has been previously been suggested. The implication is that, at least statistically speaking, it may only be necessary to simulate the counts of the largest clouds to predict the numbers of the smallest.

*Code and data availability.* Python code to analyze all data and generate all Figs. is available from the first author upon request. The VIIRS and EPIC datasets were downloaded from NASA Earthdata (NASA) and all others from the ICARE Data Center in Lille, France (ICARE).

## Appendix A: EPIC data

Due to the inaccuracy of measuring cloud perimeters and areas consisting of a small number of pixels (Christensen and Driver, 2021), we remove all clouds with perimeters $\leq 10 \times$(nadir resolution) or areas $\leq 10 \times$(nadir resolution)$^2$ (Sec. 4.3). If these same minimum thresholds are used for EPIC's cloud size distributions, results show non-power-law size distributions for both area and perimeter at the small end of the size distribution (Fig. A1). This is in contrast to all other satellite datasets over similar size ranges (Figs. 2 and 3).

As a possible explanation for this discrepancy, EPIC imagery is compressed prior to transmission to Earth by averaging $2 \times 2$ pixel regions. These regions are then interpolated back to their original resolution, artificially smoothing out the details of cloud perimeters. Since cloud perimeter lengths are resolution dependent, this results in an inaccurate perimeter measurement given EPIC's resolution. We suggest this could be a potential reason EPIC size distributions do not agree with other satellites.

It appears interpolation predominately affects measurements of area and perimeter in small clouds. To account for this inconsistency, we instead truncate EPIC's size distributions where perimeters $\leq 30 \times$(nadir resolution) or areas $\leq 1000 \times$(nadir resolution)$^2$. With these revised thresholds, results from EPIC roughly agree with those from other datasets (Figs. 2 and 3).





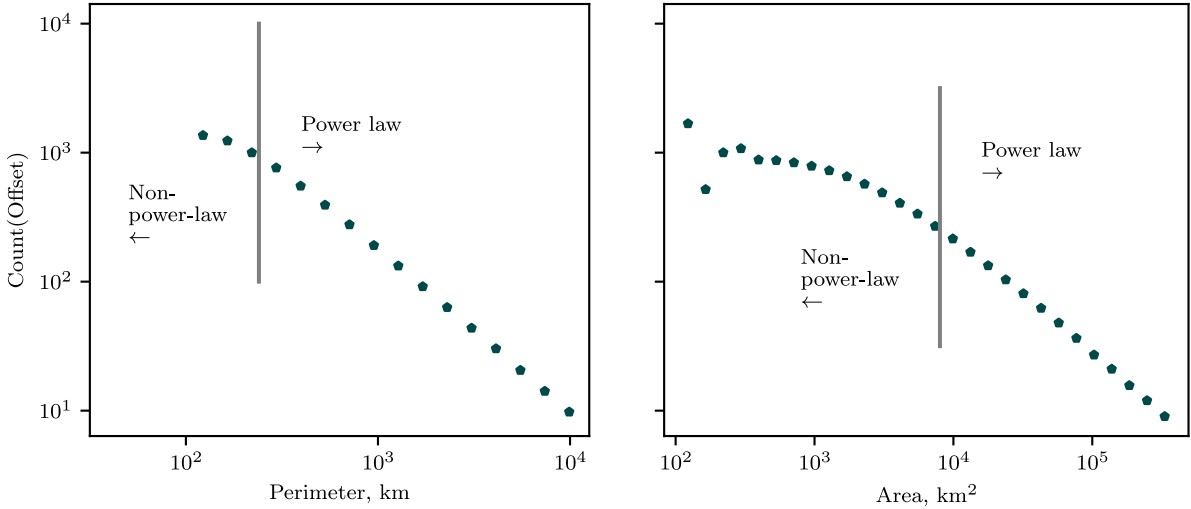

**Figure A1.** Cloud perimeter (left) and area (right) histograms for EPIC, omitting data where perimeters $\leq 10\times$(nadir resolution) or areas $\leq 10\times$(nadir resolution)$^2$. These thresholds are the same as those used for other datasets; however, EPIC displays non-power-law behavior over the range left of the grey lines. In Figs. 2 and 3, we instead use the grey lines as minimum thresholds (that is, omit perimeters $\leq 30\times$(nadir resolution) or areas $\leq 1000\times$(nadir resolution)$^2$).

*Author contributions.* Thomas DeWitt designed the data processing algorithms, processed the data, and wrote the paper. Thomas DeWitt and Timothy Garrett edited the paper with contributions from the other co-authors. All authors contributed to the interpretation of the results.

*Competing interests.* Timothy Garrett is a senior editor at Atmospheric Chemistry and Physics.

340 *Acknowledgements.*



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
