# Peer review of "Climatologically invariant scale-invariance of cloud horizontal sizes"

_EGUsphere, 2023_

## Referee Comment (RC1)

DeWitt et al. discuss here the relevance of fitting cloud size distributions, and more specifically their perimeters, by testing a power-law formula based on various satellite observations and a numerical simulation of the SAM model. They find that this power-law provides a good representation of perimeter distributions, being almost invariant across different atmospheric states. They also describe differences between satellite and model observations, probably due to the way in which clouds and their boundaries are characterized.

**General comments :**

I enjoyed reading this work by DeWitt et al., 'Observations of climatologically invariant scale-invariance describing cloud horizontal sizes ; however, I find that the study issue and the problematic of the article are not clear enough in the first three parts. I'd recommend revising the structure, especially parts 2 and 3, in order to better define the publication's issues.

My first comment concerns the introduction of the paper. While the state of the art is consequential, the motivations for cloud size analysis are missing, in particular the problems encountered by GCMs. Moreover, I find it surprising that at the end of this introduction, instead of presenting the structure of the paper, it's the results that are described.

As for parts 2 and 3, I think they could be combined into one, trying to be more direct about the motivations for studying cloud perimeters and how these have already been studied and fitted. Also, some sections of part 3 could be put in the introduction.
L131 to 156: I would have put this paragraph to the discussion about the different parameters alpha beta.

Another of my comments concerns the relevance of starting with SAM layers and comparing them with satellite data. Satellite data represents compressed data. It would have been a useful approach to compare this with the satellite data, and to show that the results are different when SAM is viewed in layers. I'd also like to see whether there is any dependence on the horizontal resolution of the model in determining the alpha and beta parameters.

It would be better to arrange references in chronological order when quoting articles (example: l.111-112).

**Minor comments:**

Figures and table:

- Figure 1: what are the x-y axis? And the labels?
- Figure 2 and 3: Be consistent with the units, use parenthesis for km$^2$ (same Fig. A1)
    Caption: what the dashed line represents ?
- Figure 3: Put « Model » as a label and deleted « satellites » in the Figure.
- Figure 5: Caption: over water (ocean) → over ocean ( right)
- I'll reverse the order of figures 4, 5 and 6 as:
    Fig.4 → Fig.5
    Fig.5 → Fig.6

Fig.6 → Fig.4

- Figure 8: what do the colors correspond to? Also, a problem in the caption «Appendix ??.»
- I don't quite understand what the insert in the middle figure does for the reader. Please be clearer,

- Table 1:

      Reverse GOES WEST and EST, METEOSAT 9 and 11

      Please make sure that sensor names are correct ( METEOSAT $\Rightarrow$ SEVIRI etc.)

Organization:

- Subsection 4.1.1 and 4.1.2 are not necessary and can put directly in the section 4.1
- L.243: You have a section 5.1 but no others.
- L.36: A more recent study ( Garett et al., 2018) → The study of Garett et al. (2018)
- Equation 3: what pmin and pmax represent? You defined 20 lines after.

- Clear-skies → cler-sky

---

## Author Comment (AC1)

We thank both Reviewer 1 and 2 for helpful comments.

In this document, italics denote reviewer comments and the grey boxes contain excerpts of the manuscript where changes were made. Blue indicates added text and  in keeping with the changes file.

In addition to minor grammatical errors and individual sentence clarifications, we slightly modified the organization of the paper at the request of Reviewer 1. We also added a paragraph in the conclusion concerning opportunities for future work:

> The distribution of cloud areas at large scales remains difficult to measure due to domain size limitations. An intriguing possibility might be to synthesize geostationary data to produce a quasi-global cloud mask product. The product would be similar to existing aerosol optical depth maps (Ceamanos et al., 2021) .

The title and abstract were updated to comply with updated ACP guidelines:

**Climatologically invariant scale-invariance  of cloud horizontal sizes**

**Abstract.**

 Cloud area distributions are a defining feature of  Earth's radiative exchanges with outer space. Cloud perimeter distributions $n(p)$ are also interesting because the shared interface between clouds and  clear-sky determines exchanges of buoyant energy and air. Here, we test using detailed model output and a wide range of satellite datasets, a first-principles  derivation that perimeter distributions follow a scale invariant power-law $n(p) \propto p^{-(1+\beta)}$ ~~, where $n(p)$ is the number density of cloud perimeters within $[p, p + dp]$ and $\beta = 1$. This value of $\beta$ was found to be in close agreement with output from a high-resolution, large eddy simulation of tropical convection. To further test this formulation, the current study evaluates $n(p)$ within near-global imagery from nine full-disk and polar-orbiting satellites. A power-law is found to apply to measurements of $n(p)$, and theobserved to besuggestingmeasured, indicating a relative scarcity of large clouds in satellite observations compared to theory and high-resolution cloud modelingthismay owe to the difference in perspective between evaluating $n(p)$ along quasi-horizontal moist isentropes rather than looking down from space. As a test of this hypothesis, numerical simulation output shows that, while $\beta \sim 1$ within isentropes, higher values of $\beta$ are reproduced for a simulated satellite view. However, the simulated value is a function of the cloud detection sensitivity where little such sensitivity is seen in satellite observations suggesting a possible misrepresentation of the physics controlling cloud sizes in simulations. A power-law also applies to satellite observations of cloud areas covering a range betweenand, a much wider range of scales than has been previously described in studies that we argue inappropriately treateda~~ the measurement domain.

The author list and contributions were also updated.

**Reviewer 1's Comments and Changes Made**

*"My first comment concerns the introduction of the paper. While the state of the art is consequential, the motivations for cloud size analysis are missing, in particular the problems encountered by GCMs. Moreover, I find it surprising that at the end of this introduction, instead of presenting the structure of the paper, it's the results that are described "*

Much of the introduction and Section 2 is dedicated to describing the theoretical background and motivation as to why cloud perimeters may exert a controlling influence on cloud field dynamics. We clarified the document structure to make clearer where motivations are described.

We agree that discussion of the structure of the paper would be helpful and added such a description at the end of the introduction.

> This paper is organized as follows. In Sect. 2, we first overview the theoretical arguments presented by Garrett et al. 2018 that led to the predicted cloud perimeter distribution. With this necessary background, prior empirical measurements of the related cloud area distribution are then discussed, along with the subtleties involved in measuring distributions of cloud sizes. The methods are presented in Sect. 3 and results from satellite observations in Sect. 4. In Sect. 5, we examine the role of perspective in measuring cloud size distributions and finally conclude in Sect. 6.

*"As for parts 2 and 3, I think they could be combined into one, trying to be more direct about the motivations for studying cloud perimeters and how these have already been studied and fitted. Also, some sections of part 3 could be put in the introduction. L131 to 156: I would have put this paragraph to the discussion about the different parameters alpha beta. "*

We agree with the review and changed the organization. The discussion of $p_{\max}$ and $p_{\min}$ was moved closer to Eqn 3 (as per another comment below). The heading "The challenge of measuring cloud size distributions" was made into a subheading, as we believe it is still a helpful demarcation in the review portion of the paper, but agree that sections 2 and 3 belong together. Section 1 was kept largely the same in the interest of keeping the perspective focused on the theory of Garrett et al., 2018, although as mentioned above we added a section describing the layout of the paper.

*"Another of my comments concerns the relevance of starting with SAM layers and comparing them with satellite data. Satellite data represents compressed data. It would have been a useful approach to compare this with the satellite data, and to show that the results are different when SAM is viewed in layers. I'd also like to see whether there is any dependence on the horizontal resolution of the model in determining the alpha and beta parameters. "*

While the compressed perspective of the model data might seem more similar to that which a satellite sees, we began by comparing the theoretical prediction ($\beta = 1$) to the satellite observations. Because this prediction relied only on the approximation $\delta h \ll \langle h^{\star} \rangle$ (line 77 in last version), it was initially surmised by Garrett et al., 2018 that $\beta = 1$ would apply to the troposphere as a whole and thus also to what was observed in satellite data. Introducing SAM layers first after the satellite analyses can be understood from this theoretical perspective. The compressed analysis then addressed the question of whether the discrepancy was due to a problem in the theory or a problem in the model.

We clarified the theoretical reasoning behind applying the theory to the troposphere as a whole. This sentence was also clarified as suggested by Reviewer 2:

> Provided the perturbation from the domain mean $\delta h$ is much smaller than the mean value $\langle h^* \rangle$, a constraint that is  satisfied even over the entire depth of the troposphere, and that

 turbulence around cloud edges is approximately isotropic (Heus et al., 2009; Heus & Jonker, 2008; Wang et al., 2009), the vertical and horizontal legs of the mixing engine are approximately the same size, so $\delta x \approx \delta z$ and $\nabla h \approx \delta h/\delta z = S$ where $S$ is the stability.

As for the dependence of $\alpha$ and $\beta$ on horizontal resolution, we agree that this is a promising avenue for further research.

*"It would be better to arrange references in chronological order when quoting articles (example: l.111-112). "*

We made this change document-wide.

*"Figure 1: what are the x-y axis? And the labels? "*

The figure mentioned shows satellite images, thus while the axes could be labelled "pixels", we did not add the label as it might be distracting. We clarified that this shows a satellite image in the figure caption.

Example RGB image, pre-processed cloud mask, and cloud masks created from various thresholds in optical reflectance $R$ for a single MODIS granule.

*"Figure 2 and 3: Be consistent with the units, use parenthesis for $km^2$ (same Fig. A1) Caption: what the dashed line represents ? "*

Changed "MODIS 250 m" to "MODIS 0.25 km" everywhere (in SI too). Changed units from e.g. "Perimeter, km" to "Perimeter (km)" everywhere and removed parenthesis in Fig. 8 axes labels to avoid confusion. We agree it was confusing that both the $\beta$ and $\alpha$ global mean lines and the Theory lines are dashed. The $\beta$ and $\alpha$ global mean lines were changed from dashed to solid.

*"Figure 3: Put Model as a label and deleted satellites in the Figure. "*

The word "model" does label the appropriate point in the Figure, and we believe the label "satellites" helpfully differentiates that point with the others, since they are different in nature, being derived from satellite datasets.

*"Figure 5: Caption: over water (ocean) $\rightarrow$ over ocean ( right) "*

We made the suggested change.

*"I'll reverse the order of figures 4, 5 and 6 as: ... "*

We made the suggested change.

*"Figure 8: what do the colors correspond to? Also, a problem in the caption Appendix ??. "*

We removed the colors in Fig. 8 as also suggested by Reviewer 2 and fixed the broken reference.

*"I don't quite understand what the insert in the middle figure does for the reader. Please be clearer, "*

We clarified that inset shows results over a larger range in $\tau$:

The middle inset displays $\beta$ vs. $\tau$ for the compressed SAM data, using a logarithmically scaled abscissa, over a larger range in $\tau$.

> *"Table 1: Reverse GOES WEST and EST, METEOSAT 9 and 11 Please make sure that sensor names are correct ( METEOSAT → SEVIRI etc.) "*

Simon Proud also commented that some of the sensor names were incorrect. The names have been corrected. "Dataset Name" was introduced as a way of differentiating between datasets. We relabeled plots and mentions of datasets in text accordingly.

> *"Subsection 4.1.1 and 4.1.2 are not necessary and can put directly in the section 4.1 "*

We removed the subsection headings 4.1.1 and 4.1.2.

> *"L.243: You have a section 5.1 but no others. "*

We divided Section 5 into a discussion of the global mean distributions vs. a discussion of the distributions in different climate regimes to provide a useful conceptual distinction, even though there is a single subheading.

> *"L.36: A more recent study ( Garett et al., 2018) → The study of Garett et al. (2018) "*

We changed this line:

> A more recent study (Garrett et al., 2018) Another study by Garrett et al., 2018 took a similar top-down approach but allowed for cloud interactions.

> *"Equation 3: what pmin and pmax represent? You defined 20 lines after "*

We moved the definition of $p_{\max}$ to be nearer Eqn 3.

> *"Clear-skies → cler-sky "*

We made this change:

> Cloud perimeter distributions $n(p)$ are also interesting because the shared interface between clouds and clear-skies across which air and buoyant energy are dissipated.clear-sky determines exchanges of buoyant energy and air.

**Reviewer 2's Comments and Changes Made**

> *"While the CRM analysis does indeed point to a reconciliation between the larger exponents in the observations and the theory of Garrett et al. (2018), the dramatic sensitivity of the CRM exponent to the cloud-defining optical depth threshold is not seen in the observations, leaving something of a puzzle remaining for future work. It would be interesting to consider using much larger domain CRMs (e.g., the DYAMOND simulations) to understand if period boundary conditions may be having an influence, or if the single simulation has some features that are representative of only a subset of cloud structures and meteorological variability. That could be left for future work. "*

We agree that the sensitivity of $\beta$ to threshold provides a puzzling disagreement between the CRM and observations. It would certainly be interesting if the periodic boundary conditions caused the sensitivity. We are already considering DYAMOND as a future path toward reconciling the disagreement, and believe both the perimeter and area distributions could provide an important metric for evaluating the realism of global CRMs.

> *"The Appendix, showing EPIC's return to consistency when potentially spurious data at the small end of the size distribution are removed, is very interesting. It suggests a warning for future satellite missions where stringent compression methods are required (e.g., stereo camera methods using very high-resolution cloud imagery). Further work is needed in this area. "*

We agree that more work is needed in this area, and are considering performing a more thorough investigation. If it proves true that the interpolation compression algorithm used in EPIC data causes the discrepancy, it would be an important finding for future satellite missions as coarse-graining and interpolating is an understandably attractive way to compress data. If the interpolation is not to blame, the fact remains that EPIC produces a very different result than all the rest of the datasets, and discovering the source of the discrepancy would be an important finding.

Discussion of the problem in the main text was slightly elaborated.

>  Unexpectedly the smaller portion of the size distributions obtained using EPIC display non-power-law behavior, in contrast to all other satellite datasets over similar scales. To ensure values for $\alpha$ and $\beta$ were calculated over only the power-law regime, thresholds used for EPIC clouds were increased to exclude perimeters $\leq 30\times$(nadir resolution) or areas $\leq 1000\times$(nadir resolution)$^2$. Conceivably, the discrepancy is caused by a compression algorithm that averages $2 \times 2$ pixel regions before data transmission. The regions are subsequently interpolated back to the original resolution, which may smooth cloud perimeters (see Appendix A for further discussion).

We also added a note in the conclusion discussing possible future work on EPIC data:

> Our results also suggest a warning for how future satellite missions are designed. The data compression algorithm used prior to transmission of EPIC data averages $2 \times 2$ pixel regions and then interpolates them back to the original resolution in post-processing. We argue this approach may produce erroneous cloud size distributions that do not follow a power-law. Further work could determine whether the interpolation adversely impacts other calculated cloud properties.

*"Line 43-44: The spurious effect of domain size on the size distribution was shown in Wood and Field (2011, e.g. Fig. 3), among others. "*

This is true. We did not make a change here, as this paragraph is intended to be a summary of the current paper, but added a brief discussion of the Wood and Field, 2011 result at line 156:

> Such a scale break would depend only on the size of domain considered, rather than some intrinsic physical property of the cloud field itself. This spurious effect of domain size has also been found to influence the measured power-law exponent for idealized 1-D cloud sizes (Wood and Field, 2011).

*"Line 47: "Suitable" for what specific aim? A better term might be "physically meaningful". "*

We made the change.

> To begin, we justify why it is  physically meaningful to look at cloud perimeters by summarizing the derivation of the cloud perimeter number distribution $n(p)$ presented by Garrett et al., 2018.

*"Line 53: Why does mixing of two air masses moisten the air? Where is the additional moisture coming from? Should this instead be "moistens the clear air outside of cloud" (which presumably is drier than the cloudy air)? Please clarify. "*

Rephrased to emphasize that we are tracking an idealized parcel around the circulation (the mixing engine), and it is this parcel that dries out and then moistens. See, for example, Figs 4 & 5 in Garrett et al., 2018.

> The foundation  follows a parcel through an idealized thermodynamic cycle around cloud edges – what was termed a "mixing engine" – defined by four "legs":
>
> 1. Moist adiabatic ascent inside cloud
>
> 2. Diabatic mixing  with clear air across cloud edge that dries the  parcel and reduces cloud perimeter
>
> 3. Dry adiabatic clear-sky descent
>
> 4. Diabatic mixing  with cloudy air across cloud edge that moistens the  parcel and lengthens cloud perimeter

> *"Line 67: "water vapor mixing ratio". Not total water mixing ratio. "*

We made the change.

> . . . and $z$, $T$, and $q$ are height, temperature, and the water vapor mixing ratio, respectively.

> *"Line 70: The full derivative (dq\*/dT), rather than the partial derivative) is appropriate here, because $T$ also depends upon p. Or is height and pressure assumed to be uniquely related (not $T$ dependent)? "*

Yes, for the purposes of calculating the saturation mixing ratio, a an accurate assumption is that the height and pressure are uniquely related. As defined in Randall, 1980, $\gamma = L/c_p \left( \frac{\partial q^\star}{\partial T} \right)_p$. A typo here was also corrected ($h' \to h^{\star\prime}$). We emphasized that the derivatives are partial derivatives:

> At a given height, perturbations in saturated static energy can be related to temperature (and hence buoyancy) perturbations $T'$ through  $h^{\star\prime} = c_p(1+\gamma)T'$ where $\gamma = L/c_p \partial q^\star/\partial T$ (Randall, 1980).

> *"Line 71: The gz term is more variable than the $T$ dependence? But $T$ also varies with height systematically. Explain why this is true only in a convectively unstable atmosphere? "*

After discussion we don't think the convectively unstable atmosphere criterion is necessary. The point being made is simply that $h^\star$ isosurfaces are roughly flat (see, for example, fig 2 in Garrett et al., 2018). We clarified this wording:

> In a  tropical atmosphere, variability in $h^\star$  between horizontal levels dominates variability within a given level, so a constant $h^\star$ surface can be approximated as lying along a surface of constant $z$ (Xu & Emanuel, 1989).

> *"Line 74-76: Provide a reference where this is derived from Fick's law (perhaps the 2018 Garrett paper). Or derive it here. "*

Fick's law here is meant as the general property that mass or energy diffusion across an interface is proportional to the energy gradient and a surface area normal to the gradient. We added a reference to Garrett, 2012, which argues for such generality (also added reference to Garrett et al., 2018 where it is applied to this problem specifically):

For a number $n_j$ of such clouds, each bin has a total cloud perimeter $n_j p_j$ and a total surface area  $\sigma = n_j p_j \delta z$. *$\sigma$ measures the component of the overall cloud surface area which is vertically oriented.* Fick's law suggests that for bin $j$, the total rate of dissipation of potential energy across cloud edge $Q_j$ due to diabatic turbulent mixing is proportional to the product of the energy gradient between cloudy and clear air $\nabla h$ and the total surface area $\sigma$ (Garrett, 2012; Garrett et al., 2018).

*"Line 78: Do you mean that turbulence is fully isotropic in circulations near cloud edges? Isotropic means that circulations have no preferential direction. I can't visualize a circulation that is isotropic. Please help. "*

We did mean isotropic turbulence. We also clarified why isotropic turbulence is useful for this case.

Provided the perturbation from the domain mean $\delta h$ is much smaller than the mean value $\langle h^* \rangle$, a constraint that is  *satisfied even over the entire depth of the* troposphere, and that  *turbulence around cloud edges is* approximately isotropic (Heus et al., 2009; Heus & Jonker, 2008; Wang et al., 2009), *the vertical and horizontal legs of the mixing engine are approximately the same size, so $\delta x \approx \delta z$ and* $\nabla h \approx \delta h / \delta z = S$ where $S$ is the stability.

*"Line 79: Dissipation does not have directional components. Can this be clarified? "*

We changed this sentence to:

Thus the  rate of dissipation of energy due to  *horizontal* mixing across cloud edge in any given size bin $j$ is ...

*"Line 87: Define pmin and pmax. Is pmin the Kolmogorov scale and pmax the circumference of the Earth? "*

We restructured the paper a little, as Reviewer 1 suggested as well, including moving the definitions of $p_{\min}$ and $p_{\max}$ nearer to Eqn. 3. It now reads as follows:

$$n(p) \equiv \frac{dn}{dp} \propto p^{-(1+\beta)}, \qquad \beta = 1, \qquad p_{\min} < p < p_{\max}. \tag{1}$$

Power-laws such as Eq. 1 are generally considered to be "scaling", since a rescaling of $p$ by some constant factor $c$ results in a constant rescaling of $n(p)$ by a constant factor $c^{-(1+\beta)}$. Of course, it is impossible for any physical system to exhibit scale invariance over an infinite range of scales, and so such scaling behavior can only be valid over a finite range $p_{\min} < p < p_{\max}$.

*"Line 99: Does the "exponential cutoff" only apply to the large end of the distribution? Are there any such constraints at the small end? "*

There may be an exponential cutoff at small scales as well, although we did not explore this as $p_{\min}$ appears to be at a scale smaller than was possible to measure using these datasets. The functional form beyond $p_{\max}$ is also unknown. We clarified which scale break is being discussed in this sentence:

> As an example, a common feature of power-law distributions describing many other natural and social systems is an exponential cutoff  (Clauset et al., 2009; Newman, 2005).

We did not add more discussion as we feel the functional form of the distribution beyond the scale breaks remains unknown.

> *"Line 103: Wood and Field (2011) provide empirical evidence of such a cut off at approximately the Rossby radius. However, they also showed that in the Tropics, where the Rossby radius is very large the cut off is actually at smaller scales than in the extratropics. "*

This is a good point, but given the caveat mentioned of larger cutoffs in the Tropics as well as the applicability of the Wood and Field, 2011 result, we did not discuss Wood and Field, 2011's measurement of the scale break. The Wood and Field, 2011 study measured cloud chords (defined as the length of a continuous string of cloudy MODIS pixels) and found characteristic maximum scales on the order of 1000's of kilometers. It was compellingly shown that domain size effects did not produce the scale break. However, the scale break may represent some characteristic size of how far the next hole in a given cloud is likely to be. If a large cloud has many holes, measured cloud chords would be "interrupted" by cloud holes. That is, the longest measured chord would be less than the longest possible chord for the cloud as a whole, the maximum distance from one end of the cloud to the other. Given, the cloud area tends to be of order the cloud length squared rather than the cloud chord squared, it is not clear how a scale break in cloud chords translates to a scale break in cloud areas.

In fact we looked deeper into this issue prior to writing this article, finding we were able to reproduce the Wood and Field, 2011 result using cloud chords, but when overall cloud length was used instead – defined as the overall distance between a clouds' westernmost and easternmost point – no scale break was found, with the largest cloud lengths approximately the size of the Earth's circumference in the tropics.

> *"Line 108: The perimeter or area power law exponent? "*

Changed to:

> For the power law exponents $\beta$ and $\alpha$, Garrett et al., 2018 found $\beta = 1.06 \pm 0.02$ in a comparison with a highly detailed numerical simulation of a tropical cloud field, in close agreement with the theoretically expected value of $\beta = 1$.

This section discusses both.

> *"Line 179: Is the varying pixel size across the swath taken into account? "*

The line mentioned describes the size of MODIS granules:

> We examine 13 tropical maritime granules, each centered between approximately 10°S and 20°N and 115°W to 140°W and covering an area approximately 1950 km wide by 2030 km long.

Yes, it is for both this calculation and perimeter/area calculations (mentioned in Sec. 3.3). The domain size calculations mentioned here were verified using geodesic calculations using the lat/lon data. All images are truncated by the requirement of limiting viewing angles to less than 60°.

> *"Line 191: Does the simulation self-aggregate as in the simulations shown in a number of different prior studies?*
>
> *Simulations might show a steady state that is or is not aggregated, and I would imagine that these would have quite different scaling properties. Can the authors comment on this? Also, I do not believe that steady state can we reached in 12 hours in these simulations. How are the authors defining steady state here? Radiative-convective equilibrium often produces a steady state, but may also have bifurcating or oscillating organization, so I would be interested in the authors' thoughts on how this may impact scaling. "*

The simulation differs from simulations that aggregate in four major ways: the horizontal wind profile is strongly sheared, the domain is relatively small, the grid size is relatively small, and the simulation includes large-scale advective tendencies of temperature and water vapor typical of strong large-scale ascent. All of these inhibit self-aggregation.

The steady-state obtained in the last 12 hours of simulation is not the steady state of RCE, which takes many days, but a steady state between the convection and the imposed large-scale advective tendencies. Fluctuations are relatively without trend over the 12 hour time period in question. The following quote is from Khairoutdinov et al., 2009 ("spin up" here refers to the first 8hr or so):

> "After the spin-up, precipitation is quasi-steady, oscillating between 9 and 13 mm day$^{-1}$. The cloud cover is in the 25-30% range. The latent and sensible heat fluxes appear to be close to a statistically steady state with a relatively small upward trend; it would be unreasonable to expect that a full equilibrium could be reached in just a few simulated hours. "

We clarified these points:

> Shallow cumulus form in the first hour of the simulation, gradually deepening into deep convection by hour  6. Beyond approximately hour 12, a steady-state  period is reached where the convection is in quasi-equilibrium with the prescibed large-scale forcing (Arakawa and Schubert, 1974; Lord and Arakawa, 1980; Lord, 1982) . During this steady-state period, the precipitation rate and cloud cover fluctuate without significant trends and the simulation does not self-aggregate (Khairoutdinov et al., 2009) .

As for the oscillating organisation, there may well be an oscillation in the simulation we used – it can be seen in cloud cover/precipitation (e.g. figs 2 and 3 Khairoutdinov et al., 2009) – with a period of about 10 hours. That we used 12 hours would imply all points of that oscillation would be included in the statistics.

In general, we interpret the "steady-state" requirement from the perspective of the theory to mean that any of the predicted statistics (e.g. the perimeter distribution) would only match observations if measurements were taken and aggregated over a sufficiently long period that any nonequilibrium state has is sampled throughout its oscillation. Having said that, some plots in Garrett et al., 2018 were made using a single timestep using this same simulation, so perhaps all states within the oscillation are compatible with the theory.

If self-aggregated simulations have a vastly different size distribution – perhaps a maximum for large aggregated clouds but very few small clouds – they are certainly incompatible with the theory, and given the robustness in the size distributions we find in this study, we would take that as strong evidence that self-aggregated simulations are nonphysical.

> "Line 199: Is the simulation also used to construct a "satellite like" projected cloud mask? "

We added a short description of this, but left most discussion in Section 5.

> After perimeters were calculated and binned for each layer, counts were summed over all layers. We also create a "satellite-like" image from the simulation, which is described in Section 5.

> "Line 228: "In the satellite observations examined here,...." "

We made this change.

> In satellite observations examined here, both cloud areas ...

> "Figure 8: What do the colors represent? They seem to be a distraction and are not discussed in the figure caption. Also, the caption refers to Appendix ??, so this needs correcting. "

We removed the colors and fixed the broken link.

> "Line 319: "$3 \times 10^5 \mathrm{km}^2$, .a scale much larger than has previously been suggested". Wood and Field (2011) showed no evidence of a scale break out to an area of $10^6 \mathrm{km}^2$, so this is not quite true. Also, there are two "beens" in the sentence, so please remove one of them. "

We made this change:

> We find, in satellite observations,  that the upper limit of scale invariance in cloud area distributions $a_{\mathrm{max}}$ has a value larger than $3 \times 10^5 \, \mathrm{km}^2$, a scale much larger than  some other studies have suggested (e.g. Cahalan and Joseph, 1989; Benner and Curry, 1998; Neggers et al., 2003; Peters et al., 2009) and close to that found in Wood and Field (2011).

**References**

Clauset, A., Shalizi, C. R., & Newman, M. E. (2009). Power-law distributions in empirical data. *SIAM review, 51*(4), 661–703.

Garrett, T. J. (2012). Modes of growth in dynamic systems. *Proceedings of the Royal Society A: Mathematical, Physical and Engineering Sciences, 468*(2145), 2532–2549. https://doi.org/10.1098/rspa.2012.0039

Garrett, T. J., Glenn, I. B., & Krueger, S. K. (2018). Thermodynamic constraints on the size distributions of tropical clouds. *Journal of Geophysical Research: Atmospheres, 123*(16), 8832–8849.

Heus, T., J. Pols, C. F., J. Jonker, H. J., A. Van den Akker, H. E., & H. Lenschow, D. (2009). Observational validation of the compensating mass flux through the shell around cumulus clouds. *Quarterly Journal of the Royal Meteorological Society: A journal of the atmospheric sciences, applied meteorology and physical oceanography, 135*(638), 101–112.

Heus, T., & Jonker, H. J. (2008). Subsiding shells around shallow cumulus clouds. *Journal of the Atmospheric Sciences, 65*(3), 1003–1018.

Khairoutdinov, M. F., Krueger, S. K., Moeng, C.-H., Bogenschutz, P. A., & Randall, D. A. (2009). Large-eddy simulation of maritime deep tropical convection. *Journal of Advances in Modeling Earth Systems, 1*(4). https://doi.org/https://doi.org/10.3894/JAMES.2009.1.15

Newman, M. E. (2005). Power laws, pareto distributions and zipf's law. *Contemporary physics, 46*(5), 323–351.

Randall, D. A. (1980). Conditional instability of the first kind upside-down. *Journal of Atmospheric Sciences, 37*(1), 125–130. https://doi.org/https://doi.org/10.1175/1520-0469(1980)037⟨0125:CIOTFK⟩2.0.CO;2

Wang, Y., Geerts, B., & French, J. (2009). Dynamics of the cumulus cloud margin: An observational study. *Journal of the atmospheric sciences, 66*(12), 3660–3677.

Wood, R., & Field, P. R. (2011). The distribution of cloud horizontal sizes. *Journal of Climate, 24*(18), 4800–4816.

Xu, K.-M., & Emanuel, K. A. (1989). Is the tropical atmosphere conditionally unstable? *Monthly Weather Review, 117*(7), 1471–1479. https://doi.org/https://doi.org/10.1175/1520-0493(1989)117⟨1471:ITTACU⟩2.0.CO;2